# Rear Earth Oxide Multilayer Deposited by Plasma Spray-Physical Vapor Deposition for Envisaged Application as Thermal/Environmental Barrier Coating

**Jie Zhong [1], Dongling Yang [1], Shuangquan Guo [1], Xiaofeng Zhang [2,*], Xinghua Liang [3] and Xi Wu [4]**

[1]  Chengdu Holy (Group) Industry Co., Ltd., Chengdu 610041, China; zhongjie@sccdholy.com (J.Z.); yanguse714@163.com (D.Y.); shuangquan2006@126.com (S.G.)
[2]  National Engineering Laboratory for Modern Materials Surface Engineering Technology & The Key Lab of Guangdong for Modern Surface Engineering Technology, Institute of New Materials, Guangdong Academy of Science, Guangzhou 510650, China
[3]  Guangxi Key Laboratory of Automobile Components and Vehicle Technology, Guangxi University of Science & Technology, Liuzhou 545006, China; lxh304@aliyun.com
[4]  School of Materials and Energy, Guangdong University of Technology, Guangzhou 510006, China; wuxi_5757@126.com
*   Correspondence: zhangXiaofeng@gdinm.com

**Abstract:** SiC fiber-reinforced SiC ceramic matrix composites ($SiC_f$/SiC CMCs) are being increasingly used in the hot sections of gas turbines because of their light weight and mechanical properties at high temperatures. The objective of this investigation was the development of a thermal/environmental barrier coating (T/EBC) composite coating system consisting of an environmental barrier coating (EBC) to protect the ceramic matrix composites from chemical attack and a thermal barrier coating (TBC) that insulates and reduces the ceramic matrix composites substrate temperature for increased lifetime. In this paper, a plasma spray-physical vapor deposition (PS-PVD) method was used to prepare multilayer Si–$HfO_2$/$Yb_2Si_2O_7$/$Yb_2SiO_5$/$Gd_2Zr_2O_7$ composite coatings on the surface of $SiC_f$/SiC ceramic matrix composites. The purpose of this study is to develop a coating with resistance to high temperatures and chemical attack. Different process parameters are adopted, and their influence on the microstructure characteristics of the coating is discussed. The water quenching thermal cycle of the coating at high temperatures was tested. The results show that the structure of the thermal/environmental barrier composite coating changes after water quenching because point defects and dislocations appear in the $Gd_2Zr_2O_7$ and $Yb_2SiO_5$ coatings. A phase transition was found to occur in the $Yb_2SiO_5$ and $Yb_2Si_2O_7$ coatings. The failure mechanism of the T/EBC composite coating is mainly spalling when the top layer penetrates cracks and cracking occurs in the interface of the Si–$HfO_2$/$Yb_2Si_2O_7$ coating.

**Keywords:** $SiC_f$/SiC CMC; PS-PVD; failure mechanism; thermal cycle; structural evolution

## 1. Introduction

With the development of aero-engines, the performance requirements for high-temperature-resistant hot-end components to resist high-temperature environmental corrosion have further increased. Due to their low density, resistance to high temperatures (up to 1650 °C), and excellent resistance to chemical attack, SiC fiber-reinforced SiC ceramic matrix composites ($SiC_f$/SiC CMCs) have become the most promising materials in the field of new-generation aero-engines. However, $SiC_f$/SiC CMC substrates are subject to rapid corrosion failure when used in aero-engines under harsh conditions such as high temperature, high pressure, and exposure to multiple molten salts. To solve this problem, thermal/environmental barrier coatings (T/EBCs) are applied to the surface of $SiC_f$/SiC CMC to improve the resistance of hot-end components to high temperatures and corrosion [1,2].

The SiC fiber-reinforced SiC ceramic matrix composites' substrate materials are effectively isolated from high-temperature corrosive environments by T/EBCs, which greatly reduces high-temperature corrosion of the materials by the environment and prolongs the service life of components. A new generation for T/EBC coatings contains Si/Mullite+BSAS/Rare-Earth (RE) silicates, which can tolerate service temperatures that exceed 1400 °C [3]. The thermal conductivity of $Gd_2Zr_2O_7$-type TBCs is considerably lower than that of 8YSZ-type TBCs, leading to their widespread application in the thermal insulation and protection of aero-engine blades [4,5]. A $Yb_2Si_2O_7$–$Yb_2SiO_5$ composite coating has thermomechanical stability and a high-temperature crack self-repair capability [6–8]. An Si–$HfO_2$ coating is intended to provide a well-bonded interface between the EBC ceramic layer and the CMC substrate, and the addition of $HfO_2$ provides better mechanical properties at higher temperatures [9].

In this paper, a plasma spray-physical vapor deposition (PS-PVD) method is used to prepare multilayer Si–$HfO_2$/$Yb_2Si_2O_7$/$Yb_2SiO_5$/$Gd_2Zr_2O_7$ T/EBC composite coatings on the surface of SiC fiber-reinforced SiC ceramic matrix composites substrates. The blades of gas turbines need EBC to protect them from corrosion in high-temperature corrosive environments, while TBC acts as thermal insulation to make them work at higher temperatures. Different process parameters are adopted, and their influence on the microstructure characteristics of the coating is discussed. To explore the failure mechanism of the coating under the service condition of a thermal cycle, the water quenching thermal cycle of the coating at high temperatures was tested. This research on new thermal environment barrier coatings in this paper will provide fundamental data and technical support for advanced civil aviation engine airworthiness.

## 2. Experimental

### 2.1. Coating Preparation

In this experiment, a PS-PVD multilayer system (Sulzer-Metco, Winterthur, Switzerland) was used for spraying experiments on a $SiC_f$/SiC CMC substrate. Each layer of powder was uniformly coated on the surface of the $SiC_f$/SiC CMC material by a spray gun (O3CP plasma gun). First, the pretreatment process was performed on the $SiC_f$/SiC CMC material. The $SiC_f$/SiC CMC material of rectangular ($25 \times 4.5 \times 4$ mm$^3$) was blasted with $Al_2O_3$ particles at 0.4 MPa (10 times). After sandblasting, the $SiC_f$/SiC CMC substrates were placed in acetone and ultrasonically cleaned for 10 min. The composite material was placed in the PS-PVD instrument after cleaning for coating by the spray treatment. Each layer of powder (Si-10%$HfO_2$, $Yb_2Si_2O_7$, $Yb_2SiO_5$, and $Gd_2Zr_2O_7$) was sprayed on the $SiC_f$/SiC CMC substrate in sequence. The spraying parameters are shown in Table 1. A schematic diagram of the finished T/EBC coating is shown in Figure 1a. The microscopic morphology of the sprayed powder in each coating is shown in Figure 1b–e. Figure 1b shows the morphology of the top layer, $Gd_2Zr_2O_7$, with particle sizes between 0.5–2 μm and irregular rhombus shape. Figure 1c shows the morphology of the middle layer, $Yb_2SiO_5$ with particle sizes in the 2–4 μm range, and irregular polygonal shapes like the $Gd_2Zr_2O_7$ powder. However, the overall sizes of the $Yb_2SiO_5$ particles are twice that of the $Gd_2Zr_2O_7$ particles. Figure 1d shows the morphology of the middle layer, $Yb_2Si_2O_7$. After pretreatment and granulation, the $Yb_2Si_2O_7$ powders were spherical particles with sizes of approximately 13 μm. The special spherical shape provides the excellent fluidity of the powder, which is conducive to its uniform flow in the spray gun [10,11]. Figure 1e shows the morphology of Si powder containing 10% $HfO_2$, with a particle diameter of approximately 40 μm. Similar to the $Gd_2Zr_2O_7$ and $Yb_2SiO_5$ powders, they are irregular polygons. However, the particle size of Si powder containing 10% $HfO_2$ is 10 and 5 times larger than $Gd_2Zr_2O_7$ and $Yb_2SiO_5$ powders, respectively.

**Table 1.** Process parameters of PS-PVD for each coating.

| Coatings | Pre-Heat Temperature (°C) | Spraying Distance (mm) | Current (A) | Ar (NLPM) | $H_2$ (NLPM) | He (NLPM) |
|---|---|---|---|---|---|---|
| Si-10%$HfO_2$ | 500 | 400 | 1650 | 110 | 6 | 0 |
| $Yb_2Si_2O_7$ | 500 | 1000 | 2600 | 100 | 0 | 20 |
| $Yb_2SiO_5$ | 500 | 1000 | 2600 | 100 | 0 | 20 |
| $Gd_2Zr_2O_7$ | 1000 | 1000 | 2600 | 35 | 0 | 60 |

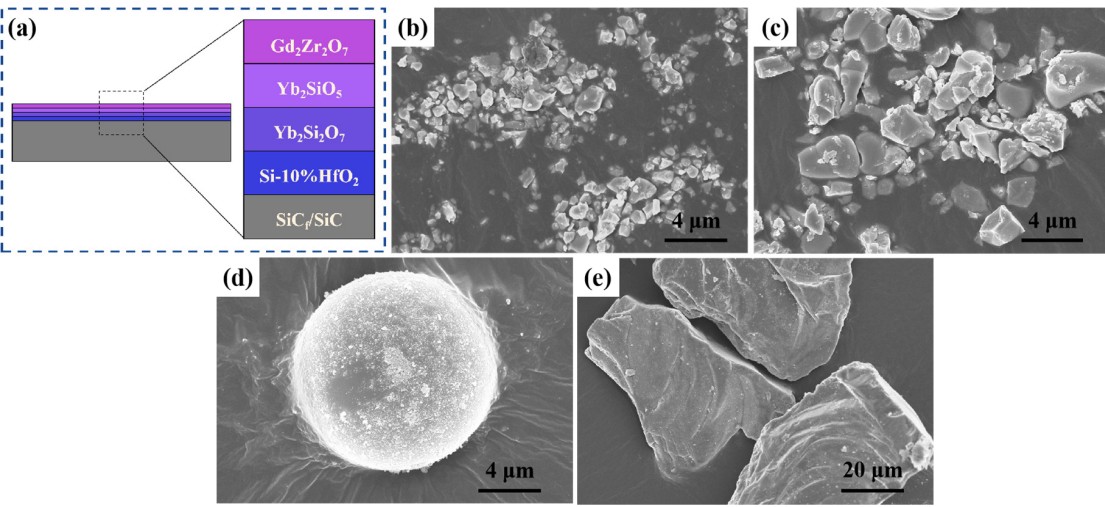

**Figure 1.** (**a**) Schematic diagram of T/EBCs coating structure. The SEM images of each coating powder of T/EBCs: (**b**) Si–$HfO_2$; (**c**) $Yb_2Si_2O_7$; (**d**) $Yb_2SiO_5$; (**e**) $Gd_2Zr_2O_7$.

### 2.2. Thermal Thermal Shock Test

Samples were subjected to a water quenching test to determine the thermal cycle performance of the T/EBC coating. The thermal cycle was divided into three stages: heating, heat preservation, and cooling. The thermal cycles consisted of 5-min heat up, 10-min dwell at 1300 °C, and water quenching for 5 min to room temperature. After 10 water quenching cycles, the microstructure of the cross-section of the sample was characterized.

### 2.3. Characterization

X-ray diffraction (XRD, D8-ADVANCE, Bruker, Hanau, Germany, Cu) analysis was performed with a radiation angle of 10°–90°, a tube voltage of 40 kV, a tube current of 40 mA, and a step length of 0.01° on the coating surface, layer by layer, to observe T/EBC surface phase formation. Field emission scanning electron microscopy (FE-SEM, ZEISS, Tholey, Germany) was used to characterize the morphology, and energy dispersive X-ray spectroscopy (EDS) was performed on the samples. To conduct an in-depth study of the cross-section of the sprayed sample, a focused ion beam (FIB, SMI3050MS2, SII, Tokyo, Japan) was used to strip the interfacial area of each coating for analysis. Transmission electron microscopy (TEM, FEI, Hillsboro, OR, USA) was used to further observe the microscopic area of the T/EBC coating. The interfacial phase was identified by selected area electron diffraction (SAED).

## 3. Results and Discussion

### 3.1. Microstructure of the As-Sprayed T/EBCs

Using the experimental parameters in Table 1, the composite T/EBC coatings were sprayed on the surface of the SiCf/SiC CMC by a PS-PVD method. The microstructure, com-

position, and elemental distribution of the T/EBC coatings are presented in Figure 2a–n. Figure 2a,b shows the micromorphology of the surface of the T/EBC coating, and an enlarged view of the microtopography of the coating surface is shown in Figure 2b. Due to the accumulation of molten powder on the surface during the spraying process, the coating surface in Figure 2b is not flat. With more porosity and holes, its surface morphology resembles the appearance of cauliflower. The EDS results for the point in Figure 2b is displayed in Figure 2c. The surface of the top layer contains only the elements O, Gd, and Zr, showing that the surface of the $Gd_2Zr_2O_7$ layer is not contaminated by other impurities. Figure 2d is the microtopography image of the cross section with T/EBCs. The cross section is smooth without cracks and pits. Figure 2e is an enlarged image of Figure 2d. The line scan result in Figure 2f shows that the cross section contains C, O, Gd, Yb, Hf, Si, and Zr, and no other impurities. Figure 2g is the general diagram of the distribution of all the elements in the coating, and Figure 2h–n is the distribution diagram of each element. Figure 2h,i shows that the Si and C contents are highest on the left side of the line scan, indicating that the base layer SiC is on the left side (Figure 2e). Figure 2h,n shows that the coating close to the base layer, SiC, has the highest content of Si and Hf, indicating that the first layer is Si–$HfO_2$. Figure 2k shows two intermediate layers containing Yb. However, it can be seen from Figure 2h,j that the Si and O contents are higher in the coating next to the Si–$HfO_2$ layer, which indicates that the second layer is $Yb_2Si_2O_7$. The third layer with less Si and O is $Yb_2SiO_5$. From Figure 2l,m, the contents of Gd and Zr are highest in the fourth layer, indicating that the top layer is $Gd_2Zr_2O_7$.

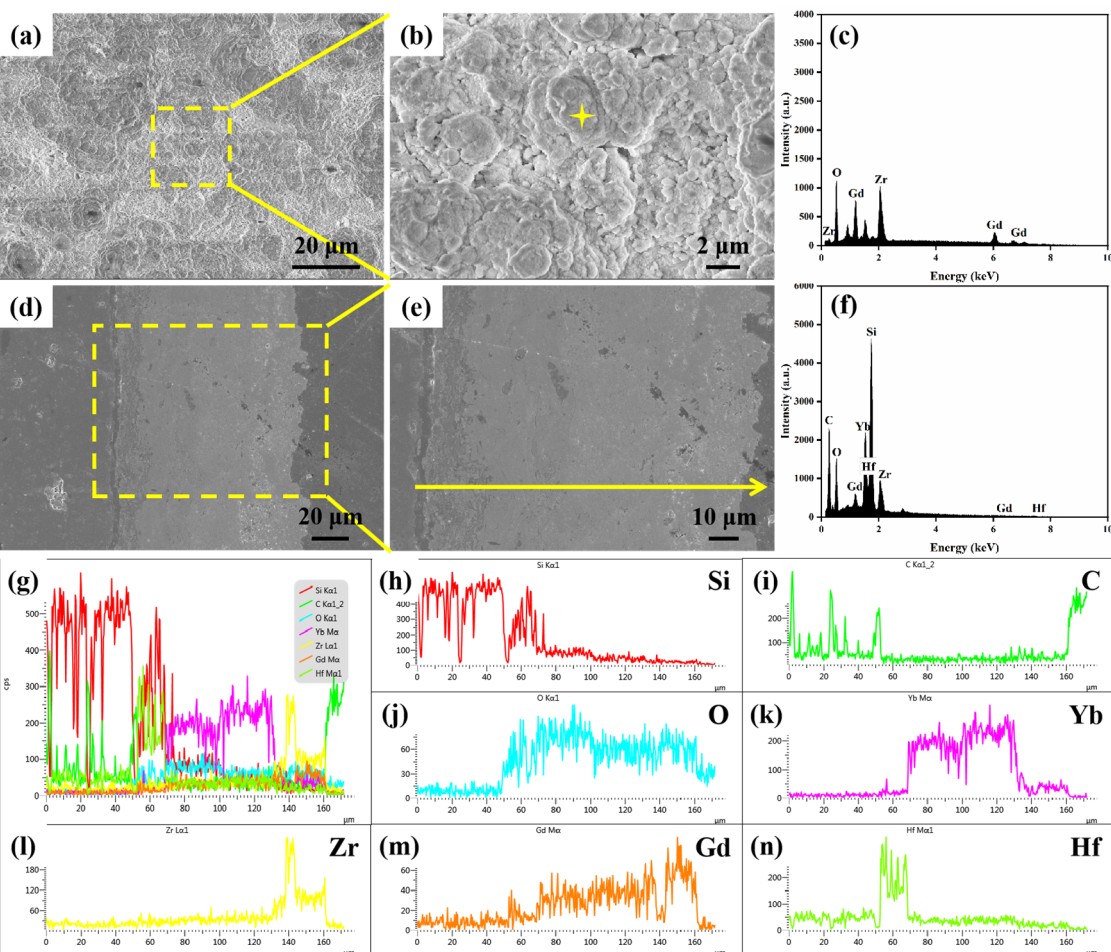

**Figure 2.** SEM images and EDS analysis of T/EBCs: (**a**) micromorphology of surface layer; (**b**) enlargement of (**a**); (**c**) point scan analysis; (**d**) micromorphology of cross-section; (**e**) enlargement of (**d**); (**f**) line scan analysis. Composition distribution of T/EBCS coatings cross section: (**g**) all elements; (**h**) Si; (**i**) C; (**j**) O; (**k**) Yb; (**l**) Zr; (**m**) Gd; (**n**) Hf.

To further investigate the phase characteristics of each coating after PS-PVD spraying, the phase structure of the coating was analyzed by X-ray diffraction. Whenever the substrate is sprayed with a new coating, small angle X-ray scattering is used for interlayer phase analysis. The XRD pattern of each coating in the T/EBCS coatings is shown in Figure 3. The positions of the diffraction peaks of each coating are consistent with the characteristic peak positions of the $Gd_2Zr_2O_7$ [12], $Yb_2SiO_5$ [13], $Yb_2Si_2O_7$ [14], and Si–$HfO_2$ [15] coatings, indicating that the coatings combined via physical stacking.

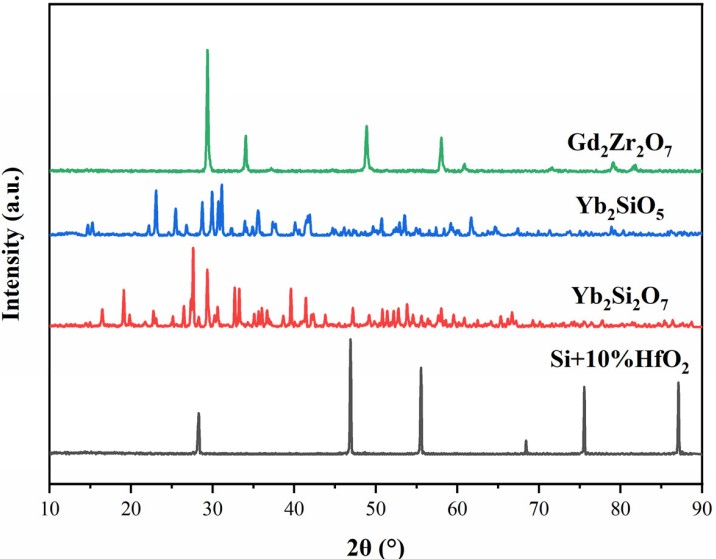

**Figure 3.** XRD patterns of each coating in T/EBCs coating.

Figure 4a shows an interfacial image of the as-sprayed T/EBC coating milled by FIB. The samples in Figure 4b–d are taken from positions 1, 2, and 3 in Figure 4a. Figure 4b shows that position 1 is the boundary of the Si–$HfO_2$/$Yb_2Si_2O_7$ coating interface, position 2 is the boundary of the $Yb_2Si_2O_7$/$Yb_2SiO_5$ coating interface (Figure 4c), and position 3 is the boundary of the $Yb_2SiO_5$/$Gd_2Zr_2O_7$ coating interface (Figure 4d).

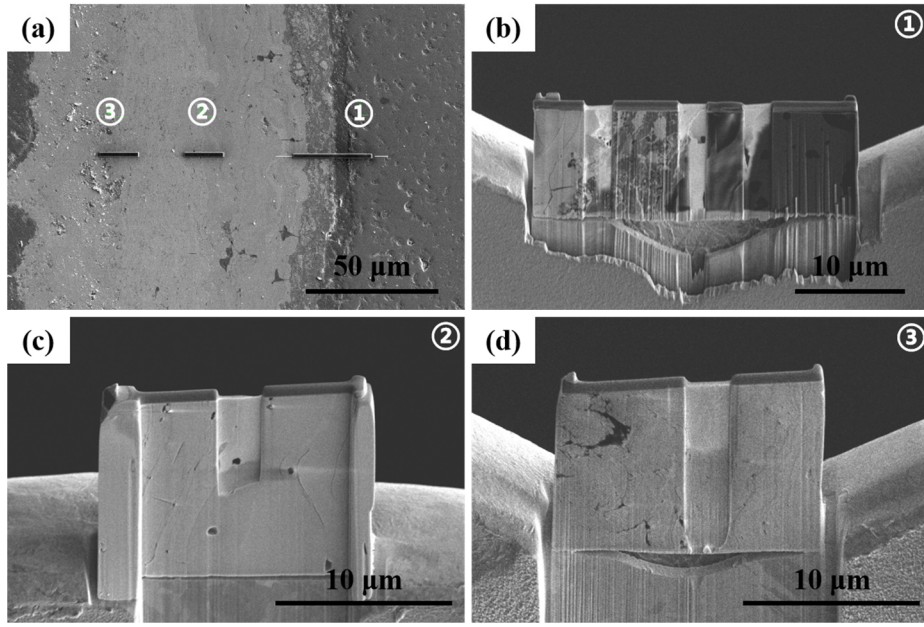

**Figure 4.** FIB-SEM of cross-section with T/EBCs coating: (**a**) cross-sectional peeling diagram; (**b**) positions 1; (**c**) positions 2; (**d**) positions 3.

### 3.2. Microstructure of the $Gd_2Zr_2O_7/Yb_2SiO_5$ Interface

The micromorphology and structure of the T/EBC coating interfaces is an important way to understand the coating performance. Figure 5a–d shows the microscopic morphology of the $Gd_2Zr_2O_7$ cross section and the structure of the $Gd_2Zr_2O_7/Yb_2SiO_5$ interface. Figure 5a shows that the coating surface is relatively dense, but there are some local pores. The top of the enlarged view (Figure 5b) resembles a feather-like cylinder. Figure 5c,d shows the $Yb_2SiO_5/Gd_2Zr_2O_7$ interface and an enlarged view. The accumulation morphology differs between $Yb_2SiO_5$ and $Gd_2Zr_2O_7$, and the interface can be clearly distinguished.

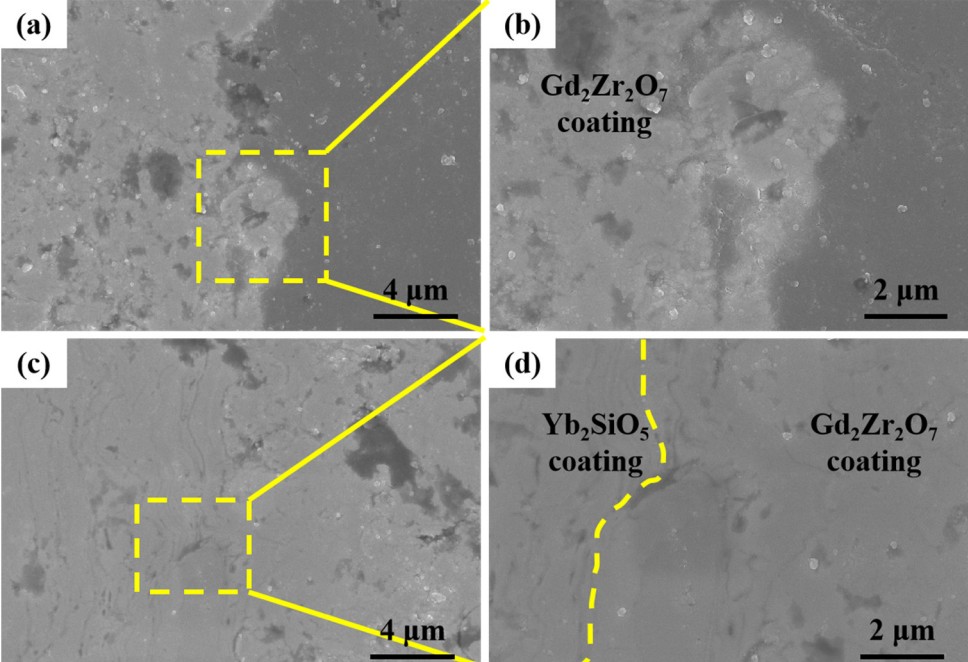

**Figure 5.** Cross-sectional micromorphology: (**a**) $Gd_2Zr_2O_7$ coatings; (**b**) enlargement of (**a**); (**c**) $Gd_2Zr_2O_7/Yb_2SiO_5$ interface; (**d**) enlargement of (**c**).

To better understand the microstructure of the $Gd_2Zr_2O_7$ coating, the coating was characterized by TEM (Figure 6), with SAED patterns and HRTEM images. Figure 6a shows that the areas marked 1 and 2 in the bright field image are the $Gd_2Zr_2O_7$ coating. The $Gd_2Zr_2O_7$ coating is characterized by diffraction rings and diffraction spots, indicating that the coatings have polycrystalline and monocrystalline phase structures. Figure 6b shows the diffraction ring of the $Gd_2Zr_2O_7$ coating, and the presence of the crystal face index (111) implies the ordered superlattice $Gd_2Zr_2O_7$ pyrochlore structure [16]. Figure 6c shows the diffraction spots of the $Gd_2Zr_2O_7$ coating. Crystal plane indices (111), (200), and (220) are mainly observed, which is consistent with the monocrystalline phase characteristics of the $Gd_2Zr_2O_7$ coating [17]. Figure 6d is the HRTEM image of the $Gd_2Zr_2O_7$ coating with defect-free structural features (no point or dislocation defects).

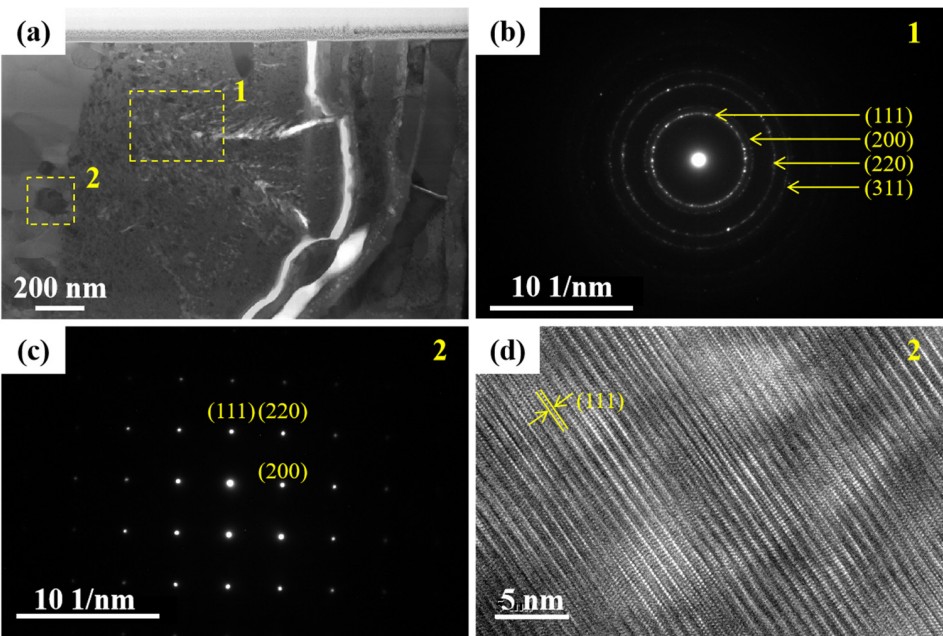

**Figure 6.** TEM image: (**a**) bright field image of $Gd_2Zr_2O_7$ coating; (**b**) the SAED patterns of $Gd_2Zr_2O_7$ coating the areas marked 1 in (**a**); (**c**) the SAED patterns of $Gd_2Zr_2O_7$ coating the areas marked 2 in (**a**); (**d**) HRTEM image of $Gd_2Zr_2O_7$ coating the areas marked 2 in (**a**).

### 3.3. Microstructure of $Yb_2SiO_5/Yb_2SiO_7$ Interface

The intermediate cross sections of the $Yb_2SiO_5$ and $Yb_2SiO_7$ coatings and their interfaces were characterized by SEM and TEM. In Figure 7a, the surface has a stacked, layered structure, with obvious stripes between layers that resemble river-like patterns [18,19]. Figure 7b is a microscopic image, and the figure shows nanoscale layers. In Figure 7c, the TEM image of the interface between the $Yb_2SiO_5$ and $Yb_2SiO_7$ coatings clearly show the two coatings and their interfaces. Figure 7d shows the diffraction spot pattern of the $Yb_2SiO_5$ coating for the $(-402)$ and $(013)$ crystal planes [20]. Figure 7e is the HRTEM image of the $Yb_2SiO_5$ coating, and it shows the $(211)$ crystal plane and crystal plane spacing of 0.387 nm, which is consistent with the XRD analysis of Figure 3. Figure 7f shows an HRTEM image of the $Yb_2SiO_7$ coating, the $(021)$ crystal plane and the crystal plane spacing of 0.319 nm, which is consistent with the XRD analysis of Figure 3. No dislocations or other defects were found in Figure 7e,f, indicating locally ordered structure in the coating.

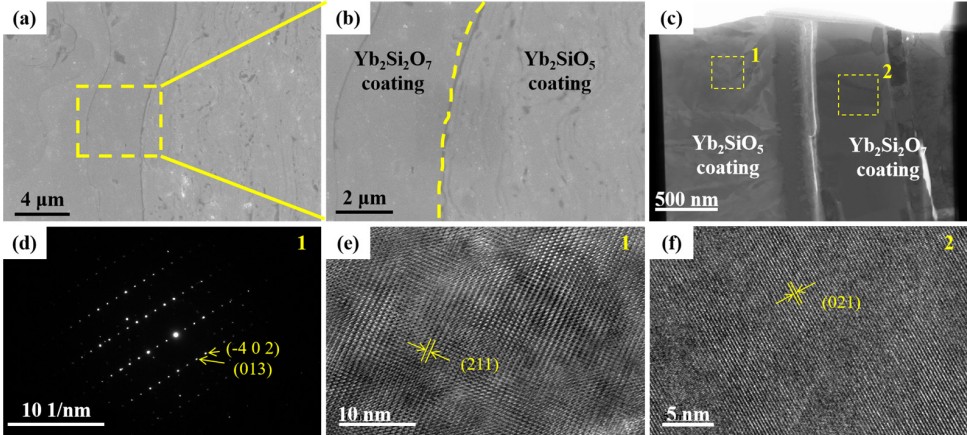

**Figure 7.** Cross-section micromorphology: (**a**) $Yb_2SiO_5/Yb_2SiO_7$ interface; (**b**) enlargement of (**a**); TEM images: (**c**) bright field image of $Yb_2SiO_5/Yb_2SiO_7$ interface; (**d**) the SAED pattern of the areas marked 1 in (**c**); (**e**) HRTEM image of the areas marked 1 in (**c**); (**f**) HRTEM image of the areas marked 2 in (**c**).

### 3.4. Microstructure of the Yb₂SiO₇/Si–HfO₂ Interface

The structural characteristics and elemental distributions of the $Yb_2SiO_7$ and $Si–HfO_2$ coatings closest to the $SiC_f/SiC$ CMC substrate were characterized by TEM and EDS. TEM and diffraction patterns of the $Yb_2SiO_7/Si–HfO_2$ interface are shown in Figure 8. The interface between the dark-colored $Yb_2SiO_7$ coating and the light-colored $Si–HfO_2$ coating is clear, and the binding is tight, without cracks. Figure 8b is an enlarged view of the light-colored $Si–HfO_2$ coating. Nano-$HfO_2$ particles are embedded in the light-colored Si coating. The HRTEM image and diffraction spot analysis of Figure 8c,d confirmed this observation [9,21]. Figure 8e–h shows the EDS patterns of the $Yb_2SiO_7/Si–HfO_2$ interface. The right side is enriched with Yb and O, indicating that the $Yb_2SiO_7$ coating is on the right side. Si is enriched on the left side, indicating that the $Si–HfO_2$ coating is on the left (Figure 8f). A small amount of interdiffusion of Yb and Si is observed at the coating interface [22].

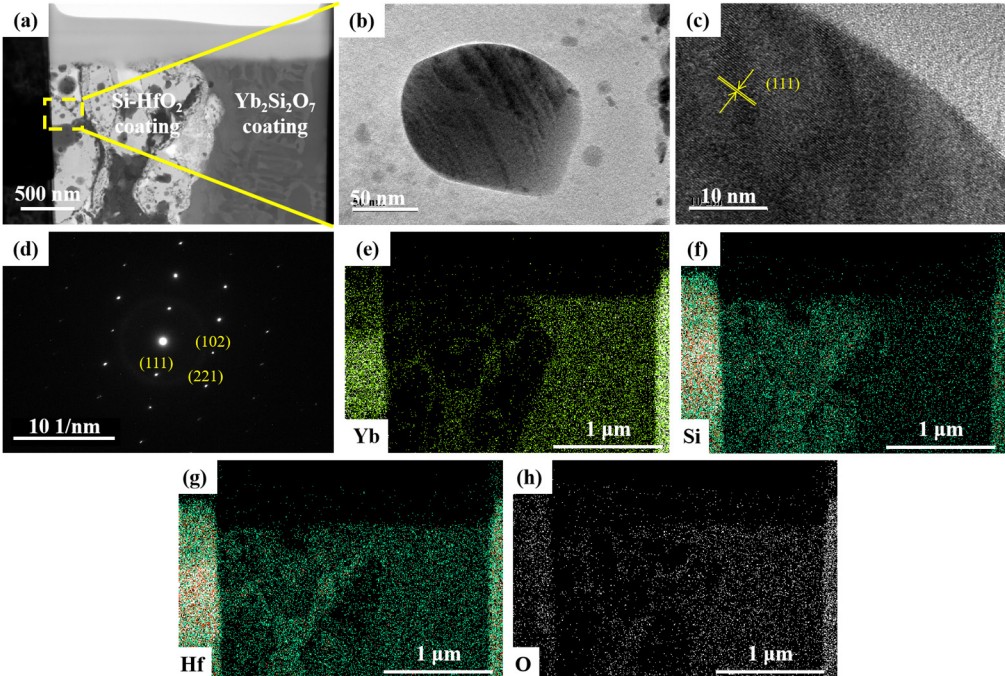

**Figure 8.** TEM image: (**a**) bright field image of $Yb_2SiO_7/Si–HfO_2$ Interface; (**b**) the TEM patterns of the $HfO_2$; (**c**) HRTEM image of the $HfO_2$; (**d**) the SAED patterns of the $HfO_2$. EDS patterns of the $Yb_2SiO_7/Si-HfO_2$ interface: (**e**) Yb; (**f**) Si; (**g**) Hf; (**h**) O.

The interfacial boundary of $Yb_2SiO_7/Si–HfO_2$ is irregular and clear (Figure 9a,b), which is consistent with the observations in Figure 8a. Figure 9c,d shows that the interface of the $Si–HfO_2$ coatings and $SiC_f/SiC$ CMC substrate is bonded continuously, and no large holes or cracks are present, which indicates that the PS-PVD technology is able to deposit a dense $Si–HfO_2$ coating on the CMC substrate. The PS-PVD technology is a hybrid coating method technology, which can prepare $Si–HfO_2$ coatings by vapor or liquid deposition [9].

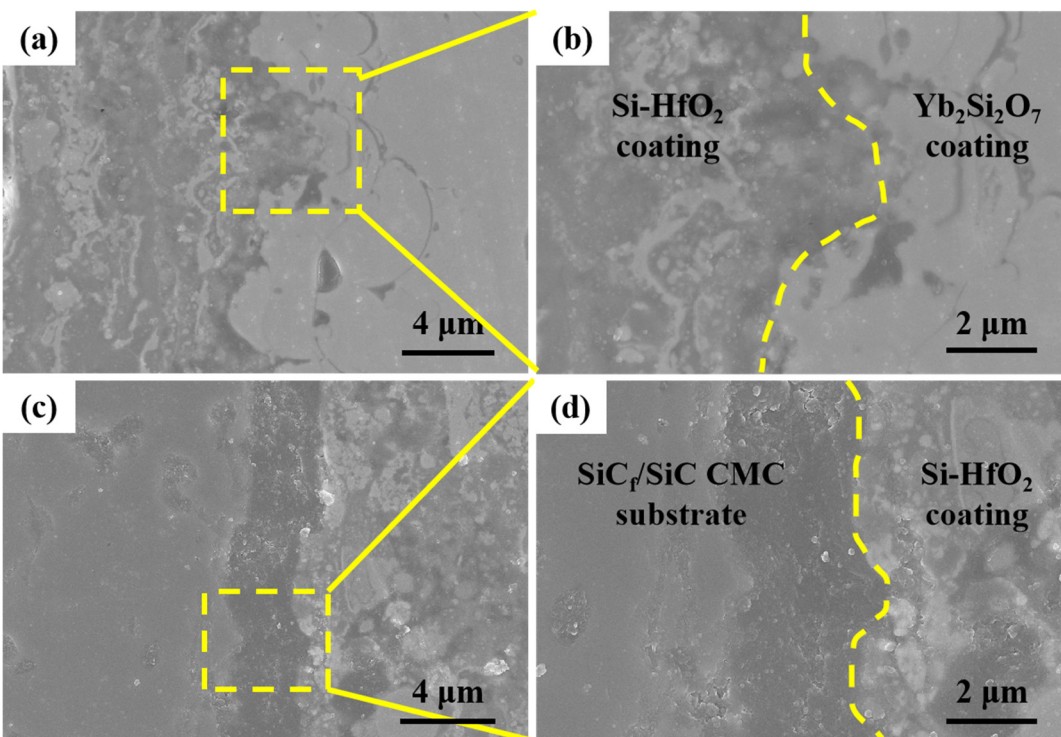

**Figure 9.** Cross-section micromorphology: (**a**) Si–HfO$_2$/Yb$_2$SiO$_7$ interface; (**b**) enlargement of (**a**); (**c**) SiC/C/Si–HfO$_2$ interface; (**d**) enlargement of (**c**).

### 3.5. Water Quenching Test and Failure Mechanism of T/EBCs

The microstructure evolution of T/EBCs was investigated by a water quenching thermal cycle at 1300 °C. The coating surface peeled off after five water quenching thermal cycles. The uppermost surface layer peeled off to half of the original coating, and the lower coating began to warp and break away from the substrate after 10 water quenching thermal cycles. Figure 10 shows the micromorphology and phase analysis of the sample after the water quenching thermal cycle test. The bond between the coating and the substrate is still intact, but penetrating cracks are found in the coating (Figure 10a). Magnifying the image shows that the penetrating cracks come from the surface layer of the coating and penetrate the Yb$_2$SiO$_7$ coating. The porosity of the Yb$_2$SiO$_5$ coating is increased (Figure 10d), and the Gd$_2$Zr$_2$O$_7$ coating is severely damaged (Figure 10b). Figure 10b,c shows two failure modes of the T/EBC coating after water quenching. The first failure mode occurs when cracks generated by the Gd$_2$Zr$_2$O$_7$ coating penetrate the Yb$_2$SiO$_5$ coating and cause failure (Figure 10b). Many point defects and dislocations were observed in the Gd$_2$Zr$_2$O$_7$ and Yb$_2$SiO$_5$ coatings (Figure 10e,f) after water quenching. These defects led to the initiation and propagation of cracks. The second mode of failure is caused by the initiation and propagation of cracks at the Si–HfO$_2$/Yb$_2$Si$_2$O$_7$ interface due to the thermal mismatch, resulting in failure. Due to the difference in the thermal expansion coefficient of Si–HfO$_2$ and Yb$_2$Si$_2$O$_7$, they are 4.5–5.5 × $10^{-6}$ K$^{-1}$ and 7.0–8.0 × $10^{-6}$ K$^{-1}$, respectively [23–26]. The composition distribution of the T/EBC coating cross sections after water quenching is shown in Figure 10g–n. Comparing the element distributions of the as-sprayed and water-quenched coatings shows that the SiC$_f$/SiC matrix and Si–HfO$_2$ matrix contain the same level of Si due to mutual diffusion, while Hf diffuses into the matrix. Gd in the top layer obviously diffuses into the T/EBC coating and Yb$_2$Si$_2$O$_7$ coating. The most obvious element is oxygen, and the oxygen content in the T/EBC coating increases obviously, which indicates that the coating underwent substantial oxidation after water quenching.

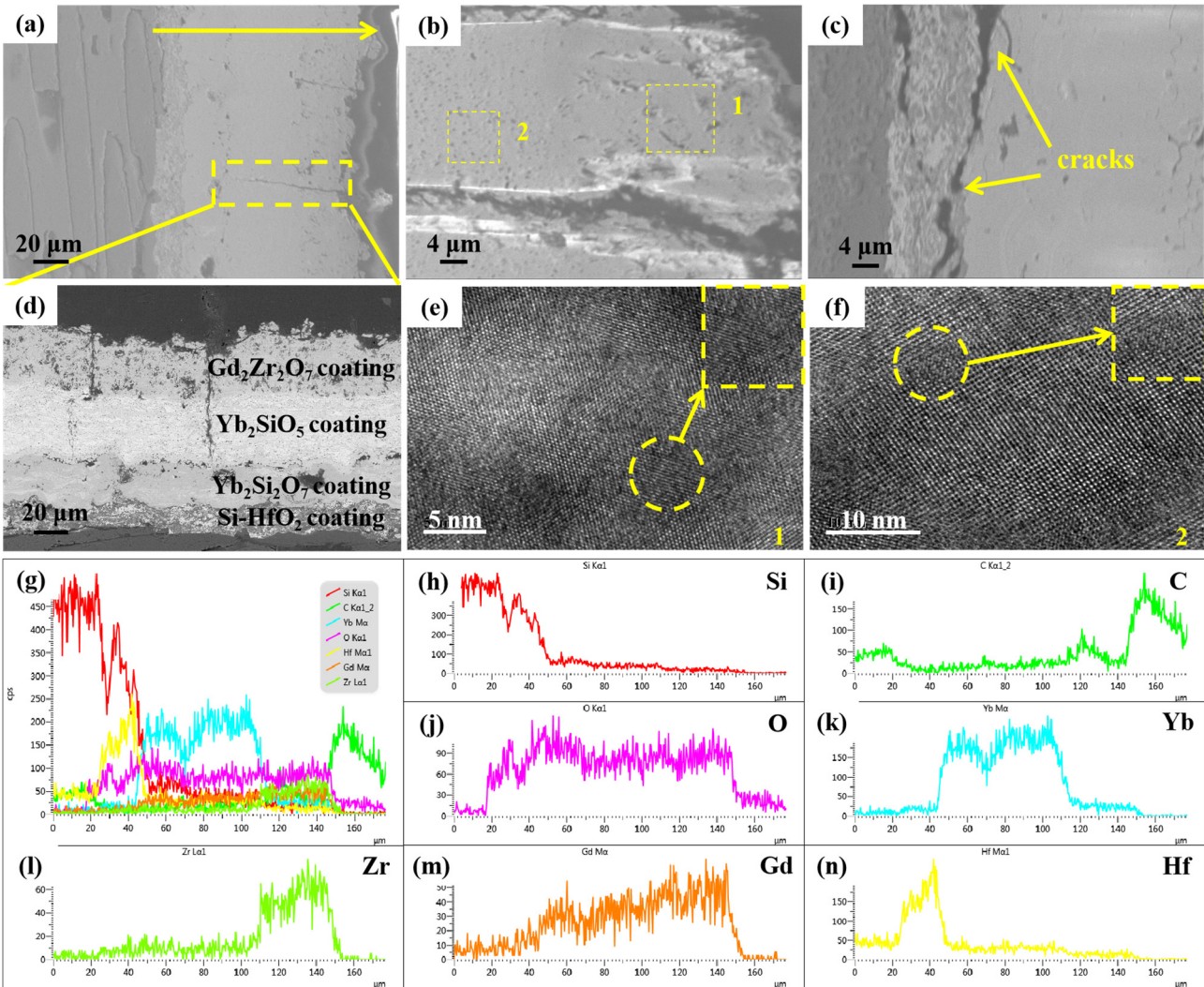

**Figure 10.** Micromorphology and phase analysis after water quenching: (**a**) cross-section; (**b**) TEM image of cracks in Gd$_2$Zr$_2$O$_7$ coatings; (**c**) TEM image of cracks in SiC/C/Si–HfO$_2$ interface; (**d**) crack enlargement of (**a**); (**e**) dislocation defect in the Gd$_2$Zr$_2$O$_7$ coating at area marked1 in (**b**); (**f**) dislocation defect in the Yb$_2$SiO$_5$ coating at area marked 2 in (**b**). Composition distribution of T/EBC$_S$ coatings cross section after water quenching: (**g**) all elements; (**h**) Si; (**i**) C; (**j**) O; (**k**) Yb; (**l**) Zr; (**m**) Gd; (**n**) Hf.

A schematic diagram of the failure mechanism with the T/EBC coating after the water quenching thermal cycle is shown in Figure 11. As seen from the figure, the failure mechanism of the coating is divided into two parts: peeling caused by the penetrating cracks on the surface, and whole layer disbonding due to the fusion of cracks between layers. The T/EBC coating was quenched in water after 10 min of the high-temperature thermal cycle at 1300 °C. The sharp temperature difference caused the growth of surface microcracks. As a result, water invaded through microcracks on the surface of Gd$_2$Zr$_2$O$_7$, conforming to the feather-like spraying morphology and reaching the Yb$_2$SiO$_5$ coating. As the amount of water quenching increased, the surface peeling became more pronounced. At the same time, cracks first grew between the Yb$_2$SiO$_7$ and Si–HfO$_2$ layers because the thermal expansion coefficients of the Yb$_2$SiO$_5$ and Yb$_2$SiO$_7$ coatings are similar. Repeated water quenching resulted in the gradual growth and fusion of interlayer cracks, forming large (4 μm) interlayer holes (Figure 10c). Related work has been reported on the water quenching test of TBC coating prepared by PS-PVD technology with the failure mechanism being surface point peeling [27], which is obviously different from this work.

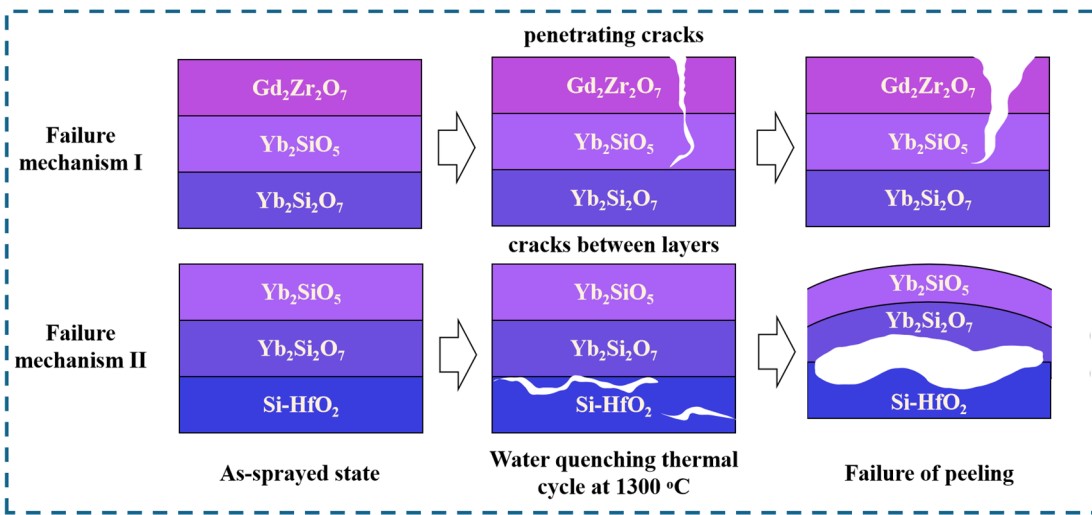

**Figure 11.** Schematic diagram of failure mechanism with T/EBCs coating after water quenching.

## 4. Conclusions

Multilayer Si–HfO$_2$/Yb$_2$Si$_2$O$_7$/Yb$_2$SiO$_5$/Gd$_2$Zr$_2$O$_7$ T/EBC composite coatings were prepared on SiC fiber-reinforced SiC ceramic matrix composites (SiC$_f$/SiC CMC) by a PS-PVD method. This study examined the effects of the coating and interface structure on the thermal cycle stability and investigated the failure mechanism. The T/EBC coatings were tested at 1300 °C with 10 cycles of water quenching. The results show the microstructure of the as-sprayed T/EBCs by the PS-PVD method, the cross section is smooth and dense without cracks and pits, for which the interlayer bonding is close, but there is no mutual diffusion of interlayer elements. Before and after water quenching, the structure of the T/EBC composite coatings differs. After water quenching, two failure modes of the T/EBC coatings were observed. One failure mode is that the cracks generated by the Gd$_2$Zr$_2$O$_7$ coating penetrated the Yb$_2$SiO$_5$ coating and caused failure. Another failure mode is caused by the initiation and propagation of cracks at the Si–HfO$_2$/Yb$_2$Si$_2$O$_7$ interface due to thermal mismatch, resulting in failure. In the next step, we will study the thermal cycle failure under different water quenching tests to obtain the performance data of this T/EBCs coating system and provide basic data and technical support for the airworthiness of advanced civil aviation engines.

**Author Contributions:** Conceptualization, validation, and writing—original draft preparation, J.Z.; investigation, software, validation, and data curation, D.Y.; methodology, project administration, and data curation, S.G.; investigation, writing—review and editing, and supervision, X.Z.; formal analysis, investigation, and visualization, X.L.; software, investigation, X.W. All authors have read and agreed to the published version of the manuscript.

**Funding:** We would like to acknowledge the financial support from the National Natural Science Foundation of China (51801034), the Guangdong Province Outstanding Youth Foundation (2021B1515020038), and Guangdong Academy of Sciences Program (2020GDASYL–20200104030), and the Guangxi Innovation Driven Development Project (No. AA18242036–2).

**Institutional Review Board Statement:** Not applicable.

**Informed Consent Statement:** Not applicable.

**Data Availability Statement:** Data is contained within the article.

**Conflicts of Interest:** The authors declare no conflict of interest.

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
