# Peer review of "Rear Earth Oxide Multilayer Deposited by Plasma Spray-Physical Vapor Deposition for Envisaged Application as Thermal/Environmental Barrier Coating"

_coatings, doi:10.3390/coatings11080889_

Round 1

Reviewer 1 Report

The manuscript concerns the T/EBC on CMCs, the manuscript is interesting, quite well prepared, sufficient explanations of the results obtained.

- Abstract. Line 15. I propose to re-edit the sentence, the abbreviation T/ECB appears earlier than it was explained.

- The Introduction is relatively short and could be improved. It is a relatively new topic but with a lot of publications. Please indicate the novelty described in this manuscript.

- Line 35 and later. Please standardize the temperature notation, additionally the temperature should be written with a space between the value and the degree sign.

- Lines 46, 48, 169 and 188. Please explain the abbreviations used. Shouldn't there be TEM instead of STEM (line 188)?

- Line 71. Please change the description of material dimensioning. 

- 3.1. Microstructure of the as-sprayed T/EBCs, lines 141-146. I was surprised by the laconic description of XRD in this section, how were the individual layers/coatings analyzed?

- Fig. 5 and description (line 164), looking at Fig. 5d, I do not see a clear interface between Yb2SiO5 and Gd2Zr2O7, on the basis of which this interface was determined?

- 4. Conclusions. How do the water quenching tests can be transfer into the real systems? Do the described damages of the coatings after 10 cycles disqualify the tested systems?

Author Response

Point 1: Abstract. Line 15. I propose to re-edit the sentence, the abbreviation T/ECB appears earlier than it was explained.

Response 1: Thank you for the comment. An explanation of the abbreviation T/EBC has been added to this sentence in the Abstract.

Point 2: The Introduction is relatively short and could be improved. It is a relatively new topic but with a lot of publications. Please indicate the novelty described in this manuscript.

Response 2: Thank you for the suggestion. The novelty described in this manuscript has been explained in the Introduction.

Point 3: Line 35 and later. Please standardize the temperature notation, additionally the temperature should be written with a space between the value and the degree sign.

Response 3: Thank you for underlining this deficiency. The notation of temperature has been standardized, and space is added between the value and the degree sign.

Point 4: Lines 46, 48, 169 and 188. Please explain the abbreviations used. Shouldn't there be TEM instead of STEM (line 188)?

Response 4: Thank you for the suggestion. The abbreviation used has been explained and STEM has been corrected to TEM.

Point 5: Line 71. Please change the description of material dimensioning. 

Response 5: Thank you for underlining this deficiency. The description of material dimensioning has been changed.

Point 6: 3.1. Microstructure of the as-sprayed T/EBCs, lines 141-146. I was surprised by the laconic description of XRD in this section, how were the individual layers/coatings analyzed?

Response 6: Thank you for the suggestion. Whenever the substrate is sprayed with a new coating, small angle x-ray scattering is used for interlayer phase analysis. The corresponding sentence has been added to the manuscript.

Point 7: Fig. 5 and description (line 164), looking at Fig. 5d, I do not see a clear interface between Yb2SiO5 and Gd2Zr2O7, on the basis of which this interface was determined?

Response 7: Thank you for the suggestion. It can be seen from Figures 5c-d that Yb2SiO5 is an obvious layered structure, while Gd2Zr2O7 is a non-layered structure. The interface between these is relatively clear.

Point 8: 4. Conclusions. How do the water quenching tests can be transfer into the real systems? Do the described damages of the coatings after 10 cycles disqualify the tested systems?

Response 8: Thank you for underlining this deficiency.In this paper, the water quenching test is a part of the simulation system, the actual system is much more complex. The purpose of thermal cycling test in this paper is to find out the failure mechanism of T/EBCs coating system under water quenching test, so as to provide improvement direction and strategy for improving the thermal cycling performance of coating. Therefore, even 10 cycles of testing is of practical significance.

Reviewer 2 Report

This manuscript is on plasma spray PVD and characterization of a rear-earth-oxide multilayer for potential application as thermal and environmental barrier coating for aircraft turbine wheel blades made of SiC composites. Unfortunately, there were found several failure effects such as cracks and defoliation of the protection layer upon the applied test procedures. With that, the positive outcome of the work is not clear. Nevertheless, a publication might be possible after revision as follows:

  1. The title is not specific and has to be changed, for example into: Rear earth oxide multilayer deposited by plasma spray PVD for envisaged application as environmental barrier coating.
  2. The abstract is not concise and includes too many abbreviations. It should be shortened and made more clear.
  3. Avoid to use so many unusual abbreviations, this makes reading hard.
  4. Can you give some more deposition details, such as energy of spray particles and growth temperature ?
  5. Try to find a more clear and specific positive outcome of the work.

Author Response

Point 1: The title is not specific and has to be changed, for example into: Rear earth oxide multilayer deposited by plasma spray PVD for envisaged application as environmental barrier coating.

Response 1: Thank you for the suggestion. We accept the suggestion and change the title to "Rear earth oxide multilayer deposited by plasma spray-physical vapor deposition for envisaged application as thermal/environmental barrier coating".

Point 2: The abstract is not concise and includes too many abbreviations. It should be shortened and made more clear.

Response 2: Thank you for the suggestion. We have revised the summary to make it more concise and clear.

Point 3: Avoid to use so many unusual abbreviations, this makes reading hard.

Response 3: Thank you for underlining this deficiency. We accept this suggestion and have modified it in this article to avoid the abuse of abbreviations.

Point 4: Can you give some more deposition details, such as energy of spray particles and growth temperature?

Response 4: Thank you for the suggestion. For the details of deposition, we have listed the process parameters of ps-pvd for each coating in Table 1. The energy and growth temperature of spray particles require special devices and instruments(such as enthalpy probe). The research work in this paper has not been measured and calculated.

Point 5: Try to find a more clear and specific positive outcome of the work.

Response 5: Thank you for the suggestion. We have modified this paper to reflect our positive results. At the same time, our research work is not perfect in the research stage of this article, and a brief description of the next research work is given.

Reviewer 3 Report

Dear Authors,

The manuscript coatings-1294301-peer-review-v1, entitled ‘Environmental/thermal barrier coatings deposited on SiCf/SiC 2 ceramic matrix composites by plasma spray-physical vapor 3 deposition’ presents the development of a composite coating system consisting of an environmental barrier coating to protect the ceramic matrix composites (CMC) from chemical attack and a thermal barrier coating that insulates and reduces the CMC substrate temperature for increased lifetime. This manuscript presents a plasma spray-physical vapor deposition (PS-PVD) method for the preparation of a multilayer Si-HfO2/Yb2Si2O7/Yb2SiO5/Gd2Zr2O7 T/EBC composite coatings on the surface of SiCf/SiC CMC. The purpose being to develop a coating with resistance to both high temperatures and chemical attack. The results presented by the authors show that the structure of the T/EBC composite coating changes after water quenching because point defects and dislocations appear in the Gd2Zr2O7 and Yb2SiO5 coatings. A phase transition was found to occur in both Yb2SiO5 and Yb2Si2O7 coatings, as shown by the authors. Moreover, the authors suggest that their research on new thermal environment barrier coatings, described in this manuscript, will provide fundamental data and technical support for advanced civil aviation engine airworthiness.

Characterization methods for surface investigation used by the authors are as follows: X-ray diffraction (XRD), field emission scanning electron microscope (SEM), energy dispersive X-ray spectroscopy (EDS), a focused ion beam (FIB), Transmission electron microscopy (TEM) and a selected area electron diffraction (SAED).

The overall manuscript text is written well, with good explanation and descriptions.

I suggest the authors to increase the image quality and the annotation fonts size for better visualization, for all the images in figure 2 ( on page 4), especially for the graphs where the lines an not so well to be discriminated by an unexperienced reader. Because it is so complex in information’s, I suggest that figure 2 to be made bigger , i.e 80-90% of the page. Figure 4 on page 5 to be made bigger ( i.e. text width), same for figure 5 on page 6. Figure 10 on page 9 to be bigger, same suggestion as for fig 2.

The conclusions are weak and should be extended to include more of the results obtained. Several further directions of study should also be included.

I propose this manuscript should be considered for publication in Coatings after meeting these suggestions.

Recommendation: MINOR Revision.

Author Response

Point 1: I suggest the authors to increase the image quality and the annotation fonts size for better visualization, for all the images in figure 2 ( on page 4), especially for the graphs where the lines an not so well to be discriminated by an unexperienced reader. Because it is so complex in information’s, I suggest that figure 2 to be made bigger , i.e 80-90% of the page. Figure 4 on page 5 to be made bigger ( i.e. text width), same for figure 5 on page 6. Figure 10 on page 9 to be bigger, same suggestion as for fig 2.

Response 1: Thank you for underlining this deficiency. The image quality and annotation fonts size of Figs. 2 and 10 have been improved for better visualization, and the sizes of Figs. 4 and 5 have also increased correspondingly.

Point 2: The conclusions are weak and should be extended to include more of the results obtained. Several further directions of study should also be included.

Response 2: Thank you for the suggestion. More results have been added to the conclusion, and the future research direction is pointed out.

Round 2

Reviewer 1 Report

The authors improved the manuscript according to the reviewer comments. Thank you very much. The manuscript may be published in present form. 

Reviewer 2 Report

The authors improved the manuscript according to the reviewer comments. Although the work shows some deficiencies of the layer system for the envisaged application, it might be published at this preliminary stage.